# Impacts of Multidrug-Resistant Pathogens and Inappropriate Initial Antibiotic Therapy on the Outcomes of Neonates with Ventilator-Associated Pneumonia

**DOI:** 10.3390/antibiotics9110760

**Published:** 2020-10-30

**Authors:** Hsiao-Chin Wang, Chen-Chu Liao, Shih-Ming Chu, Mei-Yin Lai, Hsuan-Rong Huang, Ming-Chou Chiang, Ren-Huei Fu, Jen-Fu Hsu, Ming-Horng Tsai

**Affiliations:** 1College of Medicine, Chang Gung University, Taoyuan 33305, Taiwan; cyndi0805@yahoo.com.tw (H.-C.W.); bluewing0356@msn.com (C.-C.L.); kz6479@cgmh.org.tw (S.-M.C.); lmi818@msn.com (M.-Y.L.); qbonbon@gmail.com (H.-R.H.); cmc123@cgmh.org.tw (M.-C.C.); rkenny@cgmh.org.tw (R.-H.F.); 2Division of Pediatric Neonatology, Department of Pediatrics, Chang Gung Memorial Hospital, Taoyuan 33305, Taiwan; 3Division of Pediatric Hematology/Oncology, Department of Pediatrics, Chang Gung Memorial Hospital, Taoyuan 33305, Taiwan; 4Division of Neonatology and Pediatric Hematology/Oncology, Department of Pediatrics, Chang Gung Memorial Hospital, Yunlin 638, Taiwan

**Keywords:** ventilator-associated pneumonia, respiratory failure, neonates, multidrug-resistant pathogens, broad-spectrum antibiotics

## Abstract

It is unknown whether neonatal ventilator-associated pneumonia (VAP) caused by multidrug-resistant (MDR) pathogens and inappropriate initial antibiotic treatment is associated with poor outcomes after adjusting for confounders. **Methods:** We prospectively observed all neonates with a definite diagnosis of VAP from a tertiary level neonatal intensive care unit (NICU) in Taiwan between October 2017 and March 2020. All clinical features, therapeutic interventions, and outcomes were compared between the MDR–VAP and non-MDR–VAP groups. Multivariate regression analyses were used to investigate independent risk factors for treatment failure. **Results:** Of 720 neonates who were intubated for more than 2 days, 184 had a total of 245 VAP episodes. The incidence rate of neonatal VAP was 10.1 episodes/per 1000 ventilator days. Ninety-six cases (39.2%) were caused by MDR pathogens. Neonates with MDR–VAP were more likely to receive inadequate initial antibiotic therapy (51.0% versus 4.7%; p < 0.001) and had delayed resolution of clinical symptoms (38.5% versus 25.5%; p = 0.034), although final treatment outcomes were comparable with the non-MDR–VAP group. Inappropriate initial antibiotic treatment was not significantly associated with worse outcomes. The VAP-attributable mortality rate and overall mortality rate of this cohort were 3.7% and 12.0%, respectively. Independent risk factors for treatment failure included presence of concurrent bacteremia (OR 4.83; 95% CI 2.03–11.51; p < 0.001), septic shock (OR 3.06; 95% CI 1.07–8.72; p = 0.037), neonates on high-frequency oscillatory ventilator (OR 4.10; 95% CI 1.70–9.88; p = 0.002), and underlying neurological sequelae (OR 3.35; 95% CI 1.47–7.67; p = 0.004). **Conclusions:** MDR–VAP accounted for 39.2% of all neonatal VAP in the neonatal intensive care unit (NICU), but neither inappropriate initial antibiotics nor MDR pathogens were associated with treatment failure. Neonatal VAP with concurrent bacteremia, septic shock, and underlying neurological sequelae were independently associated with final worse outcomes.

## 1. Introduction

While ventilator-associated pneumonia (VAP) is the second most common cause of healthcare-associated infections in neonatal intensive care units (NICUs), it remains challenging to accurately diagnose VAP in neonates [1,2]. Approximately 15–20% of premature neonates with intubation for more than two days experience at least one episode of VAP during hospitalization [1,2,3,4] and the mortality rate of neonatal VAP is reported to be around 9.3–16.4% [5,6,7]. In contrast to coagulase-negative staphylococcus (CoNS) being the most common cause of neonatal late-onset sepsis, Gram-negative bacilli account for the majority of neonatal VAP [6,7,8,9]. As clinicians tend to use broad-spectrum antibiotics when critically ill neonates have clinical deterioration [8,9], these patients are likely to have antibiotic overuse. Therefore, antibiotic-resistant pathogens are emerging after antibiotic selection, and neonates face an increased risk of antibiotic-associated composite outcomes [10,11,12].

There have been various studies of antibiotic stewardship programs in the NICU in order to promote the appropriate use of antibiotics for neonates [12,13,14]. Both Infectious Diseases Society of America and American Academy of Pediatrics guidelines recommend restricting the use of broad-spectrum antibiotics to neonates with a high risk of meningitis or those present with respiratory failure and/or septic shock [15,16]. However, justified broad-spectrum empirical antibiotics are often used by clinicians in neonates with high-risk of resistance, i.e., previous antibiotic exposure or endotracheal *Pseudomonas* genus colonization [7,17]. Furthermore, healthcare-associated infections (HAIs) caused by multidrug-resistant (MDR) pathogens are becoming alarmingly frequent in the NICU, accounting for nearly one-fifth of all nosocomial infections, and they are associated with higher mortality and morbidity [18,19,20]. Currently, there are no standard guidelines for empiric antibiotic use in neonates with clinically suspected VAP, and the impact of inappropriate initial antibiotic use on the final outcome has not been investigated. Therefore, we aim to examine empiric antibiotic administration for neonatal VAP and the impacts of MDR pathogens on the outcomes.

## 2. Methods

### 2.1. Patients, Study Design, and Setting

We prospectively observed all neonates who had clinical symptoms and signs of VAP, with mechanical intubation for more than 2 days, in the NICUs of Chang Gung Memorial Hospital (CGMH) between October 2017 and March 2020. There are three NICUs at CGMH, with a total capacity of 49 beds equipped with ventilators and 58 beds of special care nurseries. This study is part of our research project that investigated the application of nonbronchoscopic bronchoalveolar lavage (NBBAL) for the diagnosis of neonatal VAP. Therefore, all neonates with clinically suspected VAP underwent standard endotracheal aspiration for the culture of the aspirate samples, and some of them had an NBBAL examination [21]. The NBBAL technique has been proven to be a useful alternative to traditional BAL and can be used in extremely preterm neonates [21,22]. We considered repeated isolates in the same patient and the same episode of VAP as one case. For the complete treatment of the previous case, a new VAP episode in the same patient was considered as a separate case and was defined according to standard neonatal VAP diagnostic criteria. This study was approved by the Institutional Review Board of CGMH, and written informed consent was obtained from parents of the neonates prior to being enrolled in this study. 

### 2.2. Definition

We applied the updated diagnostic criteria of the Centers for Disease Control and Prevention (CDC) for neonatal VAP (Appendix A
Table A1), which defines neonatal VAP as a nosocomial infection diagnosed in patients undergoing mechanical ventilation for at least 48 hours [1,23]. Therefore, all clinical, radiological and laboratory, and microbiological criteria were prospectively observed to fulfill the diagnostic criteria of neonatal VAP. As the CDC permits the diagnosis of “clinically defined pneumonia” based only on clinical and radiological findings without any isolated pathogen [1,23], we enrolled neonates who did not receive NBBAL and only sputum cultures were available.

Limited-spectrum antibiotic treatments are defined as the use of ampicillin/sulbactam, oxacillin, ceftriaxone, or gentamicin [24]. Resistance to limited-spectrum antimicrobial therapy was considered when a bacterial strain was resistant to one or more of these antibiotics. For VAP with more than one strain, resistance was considered when one bacteria species was not susceptible to limited-spectrum antimicrobial therapy. Antibiotic susceptibility patterns were determined according to methods recommended by the National Committee for Clinical Laboratory Standards Institute (CLSI) for the disk diffusion method, and a categorical assignment was performed using CLSI breakpoints [25]. Multidrug-resistant (MDR) pathogens were defined as those resistant to at least one agent in three or more antimicrobial categories: carbapenems (imipenem and meropenem), penicillins (piperacillin, ticarcillin, and piperacillin/tazobactam), broad-spectrum cephalosporins (ceftazidime and cefepime), monobactams (aztreonam), aminoglycosides, and fluorquinolones [20,26]. In cases of polymicrobial VAP, which was defined as more than one pathogenic microorganism identified from a single set of sputum culture, the episode was defined as an MDR case if one of the isolates was an MDR pathogenic strain.

All comorbidities of prematurity, including respiratory distress syndrome, intraventricular hemorrhage, bronchopulmonary dysplasia (BPD), necrotizing enterocolitis, and periventricular leukomalacia, were based on the latest updated diagnostic criteria in the standard textbook of neonatology [27]. Inappropriate empirical antibiotics were considered when one of the bacteria strains was resistant to empirical treatment. Antibiotic exposure was defined as the systemic administration of an antibiotic class for at least 3 days in the preceding 30 days before the onset of VAP.

### 2.3. Data Collection

Patient demographics, clinical parameters at VAP onset, therapeutic interventions, responses to treatment, and outcomes were prospectively collected for all neonates with clinically diagnosed VAP. The timing of performing chest X-ray imaging and obtaining endotracheal aspirates or NBBAL was defined as the VAP onset. The severity of illness was evaluated at the onset of each VAP episode using the neonatal therapeutic intervention scoring system (NTISS) [28]. All patients were followed until discharge or death. The case was excluded if the patient was transferred to other hospitals and the final outcome was unknown.

### 2.4. Statistical Analysis

Variables with parametric distributions are expressed as mean (standard deviation, SD), and continuous variables with nonparametric distributions are expressed as median (interquartile range, IQR). Comparisons between continuous variables of different subgroups were analyzed using paired Student’s t-tests and paired Wilcoxon rank-sum tests. Categorical variables were compared with chi-square tests or Fisher’s exact tests. All p-values were two-tailed, and p-values < 0.05 were considered to be statistically significant. All statistical analysis was performed using SPSS (version 21.0; IBM, Armonk, NY, USA).

The culture of NBBAL fluid was considered positive if a potential pathogen was isolated, regardless of the number of CFUs per milliliter. Subgroup analyses were performed with MDR pathogens and antibiotic-susceptible pathogens-associated VAP, the empirical use of broad-spectrum or limited spectrum antibiotics, and initial appropriate vs. inappropriate antibiotic treatment. The primary outcomes were VAP-attributable mortality and final in-hospital mortality. As we aimed to investigate the impacts of therapeutic antibiotics on the outcomes, the secondary outcome was treatment failure of the VAP episodes. Treatment failure of VAP included neonates who died directly due to VAP, those who required therapeutic antibiotics for more than two weeks, progression to bacteremia, and clinical deterioration occurring after 7-days of effective antibiotic treatment. Therefore, risk factors of treatment failure in neonatal VAP were identified using univariate and multivariate logistic regression analyses. All variables with *p*-values < 0.1 were enrolled in the multivariate logistic regression model.

### 2.5. Availability of Data and Materials

The datasets used/or analyzed during the current study are available from the corresponding author upon reasonable request.

### 2.6. Ethics Approval and Consent to Participate

This study was approved by the institutional review board of Chang Gung Memorial Hospital, and written informed consent was obtained from the parents of the neonates prior to being enrolled in this study.

## 3. Results

### 3.1. Epidemiology of VAP and the Microorganisms

During the study period, a total of 720 preterm and/or critically ill neonates had mechanical intubation for more than 2 days and were prospectively observed for a total of 24,281 neonates–ventilator days. Among them, a total of 245 episodes in 184 neonates met the criteria of VAP. All VAP cases were included in the analysis because all the patients were followed until discharge or death. The median (IQR, interquartile range) gestational age and birth weight of our cohort were 26.0 (25.0–29.0) weeks and 882.0 (715.0–1154.0) g, respectively. The incidence rate of neonatal VAP in our cohort was 10.1 episodes/per 1000 neonates–ventilator days. The median (IQR) time of VAP onset was 29 (19–48) days of life.

The respiratory tract specimens were primarily sampled by endotracheal aspiration (66.9%) and NBBAL (33.1%). All the pathogens are listed in Table 1. Of the 297 strains of microorganisms identified as pathogens of 245 episodes of VAP, 207 (69.7%) were Gram-negative bacilli and 85 (28.6%) were Gram-positive cocci. The most common pathogenic microorganism spectra of neonatal VAP were *Staphylococcus aureus* (82 [27.6%]), followed by *K. pneumonia* (43 [14.5%]), *Pseudomonas aeruginosa* (33 [11.1%]), *Acinetobacter baumannii* (31 [10.4%]), *Serratia marcescens* (21 [7.1%]), and *E. coli* (21 [7.1%]). In the 245 episodes of VAP, 96 (39.2%) were caused by MDR pathogens, and 46 (18.8%) were accompanied by concurrent bacteremia (Table 1). A total of 5 fungal strains were identified, and all these *Candida* isolates were interpreted as colonization.

### 3.2. Comparisons between MDR–VAP and Non-MDR–VAP

The case demographics, underlying chronic comorbidities, and clinical features of all VAP episodes are presented in Table 2. Eighty-nine (36.3%) of these VAP episodes occurred in neonates on a high-frequency oscillatory ventilator (HFOV), and 39 (15.9%) had septic shock requiring cardiac inotropic agents and/or intravascular volume expansion. The MDR pathogen-associated VAP (MDR–VAP) did not significantly differ from non-MDR pathogen-associated VAP episodes (non-MDR–VAP) in terms of patient demographics, clinical features, or severity of illness (by NTISS scores at the onset of VAP) (Table 2). Most cases (203/245, 82.9%) had underlying chronic comorbidities, and 44.1% (*n* = 108) had two or more chronic comorbidities. Furthermore, 35.9% (*n* = 88) of VAP episodes occurred in neonates who were on antibiotic treatment for a previous episode of nosocomial infections.

### 3.3. Therapeutic Outcomes and Impacts of Inappropriate Initial Antibiotics

All these VAP episodes were treated with empiric antibiotics, and 79.6% of them were treated with empiric broad-spectrum antibiotics, namely, carbapenem or ceftazidime or cefotaxime plus vancomycin or teicoplanin (Table 3). Patients with MDR–VAP more frequently received inappropriate initial antibiotic treatment when compared with the susceptible control group. After sputum culture and antimicrobial susceptibility testing results, 156 (63.7%) episodes of VAP had a modification of their therapeutic antibiotics. The median (IQR) treatment duration of all VAP episodes was 9.8 (7.5–12.0) days. A total of 39 VAP episodes were considered as treatment failure, including death after VAP (*n* = 9), death due to superinfection after VAP (*n* = 2), progression to bacteremia (*n* = 11), requirement of therapeutic antibiotics for more than two weeks (*n* = 19), and worsening of clinical symptoms after appropriate antibiotics for more than one week (*n* = 11). The VAP-attributable mortality rate and final in-hospital mortality rate of this cohort were 3.7% (9/245) and 12.0% (22/184), respectively. MDR–VAP episodes were more likely to have delayed resolution of clinical symptoms (38.5% versus 25.5%; *p* = 0.034), were more often treated with broad-spectrum antibiotics, and were treated for a longer duration (11.0 ± 3.9 vs. 9.9 ± 3.0 days; *p* = 0.020), although the VAP attributable mortality rate and overall treatment outcomes were comparable with the non-MDR–VAP episodes.

Results of univariate and multivariate analyses of factors potentially associated with treatment failure in VAP are summarized in Table 4. Lower birth weight and extremely preterm infants were not independently associated with treatment failure. Treatment failure was not independently associated with antibiotic-resistant pathogens, inappropriate initial antibiotic treatment, or any specific pathogens. After adjustment, independent risk factors for treatment failure in VAP were presence of septic shock (OR 3.06; 95% CI 1.07–8.72; *p* = 0.037), neonates on HFOV (OR 4.10; 95% CI 1.70–9.88; *p* = 0.002), patients with underlying neurological sequelae (OR 3.35; 95% CI 1.47–7.67; *p* = 0.004), and concurrent bacteremia (OR 4.83; 95% CI 2.03–11.51; *p* < 0.001). The goodness-of-fit test of Hosmer and Lemeshow showed good agreement between the observed and predicted values of the model (*p* = 0.82).

## 4. Discussion

This is the largest prospective clinical study on the clinical features, pathogens, and impacts of empiric antibiotics on the outcomes of neonates with VAP. In this series, most neonates with VAP were extremely preterm (less than 28 weeks), had previous nosocomial infections, and had at least one chronic comorbidity. The VAP-attributable mortality rate and overall mortality rate of this cohort were 3.7% and 12.0%, respectively, which are consistent with previous studies [1,7]. MDR pathogens accounted for 39.2% of all VAP episodes, but neither antibiotic-resistant pathogens nor inappropriate initial antibiotics were associated with treatment failure. The independent risk factors of treatment failure included the presence of concurrent bacteremia, septic shock, neonates on HFOV, and underlying neurological sequelae.

Current guidelines recommend that high-risk patients with potential for septic shock, clinical deterioration, and organ dysfunction warrant emergent broad-spectrum antibiotics [29,30]. In our cohort, these patients were at high risk of antibiotic-resistant VAP due to previous antibiotic exposure, having CVC in situ, presence of chronic comorbidities, and previous HAIs. All these factors have been documented as risk factors of antibiotic-resistant HAIs [20,26,31,32,33]. Furthermore, 89 (36.3%) of VAP occurred in neonates on HFOV treatment, 75% had severe illness (NTISS >25) [28], and 39 (16.0%) had septic shock. Therefore, empirical broad-spectrum antibiotics were frequently prescribed. Although we cannot conclude whether clinical deterioration or organ dysfunction would have happened if limited-spectrum antibiotics were used at the onset of VAP, we found inappropriate initial antibiotics did not significantly affect the outcomes. It seems that VAP in neonates does not rapidly progress to severe sepsis or meningitis if the patient does not receive adequate antibiotic treatment immediately. Therefore, further studies of randomized controlled trials to develop antibiotic stewardship programs for neonatal VAP are still warranted to avoid the overuse of broad-spectrum antibiotics [14,34].

The pathogen distributions in our series were generally consistent with previous studies [1,7,35,36], although MDR pathogens were significantly higher. Our previous data showed that MDR Gram-negative bacteremia (GNB) accounted for 18.6% of all GNB bacteremia, and methicillin-resistant *S. aureus* accounted for 11.3% of all neonatal late-onset sepsis [20,37]. As CoNS remains the most common pathogen of neonatal late-onset sepsis [6,37], vancomycin or teicoplanin is frequently prescribed in the NICU. In addition, most of our neonates were extremely preterm and had received antibiotic coverage for GNB since birth. Therefore, it is reasonable that MDR Gram-negative bacilli will survive after antibiotic selection [12], being more likely to colonize the endotracheal tube and emerge as the primary pathogens of VAP in the NICU. Interestingly, most *Pseudomonas aeruginosa* strains from endotracheal aspirates were not MDR pathogens, but most of the *E coli* and *K. pneumonia* strains were multidrug-resistant. Therefore, we suggest that environmental surveillance of NICUs include epidemiologic factors, which can be used to develop decision-support models for empiric antibiotic choices in the NICU [38].

The therapeutic results of VAP were often complicated because other coinfections and subsequent episodes of HAIs often occur. For example, a catheter tip culture had a positive growth of CoNS while the patient was on ceftazidime treatment for VAP. In our cohort, most patients had a catheter in situ, and some of them had other artificial devices or surgical risk factors. The underlying chronic comorbidities further enhanced the possibilities of multiple infectious foci or superinfections. All these factors accounted for nearly two-thirds of all VAP episodes that had a modification of therapeutic antibiotics, even though initial appropriate antibiotics had been prescribed. Therefore, neonates who required antibiotic treatment for more than two weeks and those who had deterioration of clinical conditions after one week of appropriate antibiotic treatment were considered as treatment failure. 

Although it remains challenging to diagnose neonatal VAP in the NICU, we applied the most widely recognized diagnostic criteria of neonatal VAP and followed these cases prospectively. Seven (2.9%) of the VAP episodes showed no isolated pathogen in the sputum culture, but these cases were still considered definite VAP because they fulfilled all the clinical and radiological criteria of neonatal VAP and these patients were on antibiotic treatment. Furthermore, the incidence rate of neonatal VAP in this study was compatible with the literature, ranging from 2.7–10.9 episodes/1000 mechanical ventilation days in developed countries [1,2,35,36], and was significantly lower than the incidence rate of 32–37.2 episodes/1000 ventilator days in developing countries [1]. Previous studies have documented that endotracheal aspiration with nonquantitative culture of the aspirate-to-diagnosis VAP is similar to BAL with the quantitative culture of BAL fluid in clinical outcomes and antibiotic use of VAP [39]. This conclusion was also proven in pediatric ICU [40,41], although none have been conducted in the NICU [42]. Therefore, we will continue our current study to investigate the diagnostic accuracy of endotracheal aspiration for neonatal VAP compared with NBBAL.

There are some limitations to this study. The generalizability of this study is limited due to the single-center design, and the therapeutic policies depend on the attending physicians of our institute. Because of the significantly higher severity of illness in tertiary level NICUs, the antibiotic-resistant pathogens are potentially more prevalent, and our attending physicians tend to use broad-spectrum antibiotics more often. Therefore, these results are less applicable to nonteaching hospitals. Second, the sample size of this study is not adequate to report these results with high confidence. Furthermore, this is not a randomized trial, and the clinical decision was involved in determining the classification of VAP, potentially affecting the treatment outcomes. Strengths of our study include our application of the most widely recognized diagnostic criteria, no missing data due to the prospective study design, and complete follow-up of all subjects.

## 5. Conclusions

In conclusion, MDR–VAP accounted for nearly two-fifths of all VAP episodes and was more likely to be initially treated with inappropriate antibiotics. Although final treatment outcomes were comparable between MDR–VAP and non-MDR–VAP, neonates with MDR–VAP had a longer duration of antibiotic treatment and delayed resolution of clinical symptoms. Inappropriate initial antibiotic treatment did not significantly affect the outcomes. However, further studies to develop antibiotic stewardship programs for neonatal VAP are warranted to avoid the overuse of broad-spectrum antibiotics. Lastly, more aggressive therapeutic policies are required for neonatal VAP with concurrent bacteremia, septic shock, and underlying neurological sequelae in order to optimize the outcomes.

## Figures and Tables

**Table 1 antibiotics-09-00760-t001:** Pathogen distribution of neonatal ventilator-associated pneumonia (VAP) in the neonatal intensive care unit (NICU) of Chang Gung Memorial Hospital (CGMH), October 2017 to March 2020.

	All VAP Episodes (total *n* = 245)	Multidrug Resistant Pathogens Associated VAP (total *n* = 96)	VAP with Concurrent Bacteremia (Total *n* = 46)
Gram-positive cocci	46 (18.8)	15 (15.6)	5 (10.9)
*Methicillin resistant Staphylococcus aureus*	15 (6.5)	15 (15.6)	3 (6.5)
*Methicillin sensitive Staphylococcus aureus*	28 (11.4)	0 (0)	2 (4.3)
Enterococcus spp.	1 (0.4)	0 (0)	0 (0)
*Group B Streptococcus*	2 (0.8)	0 (0)	0 (0)
Gram-negative bacilli	120 (49.0)	47 (49.0)	30 (65.2)
*Pseudomonas aeruginosa*	21 (8.6)	3 (3.2)	6 (13.0)
*Escherichia coli*	15 (6.1)	12 (12.6)	5 (10.9)
*Klebsiella pneumonia*	20 (11.8)	13 (13.7)	7 (15.2)
*Klebsiella oxytoca*	3 (1.2)	0 (0)	0 (0)
*Klebsiella aerogenes*	6 (2.4)	4 (4.2)	2 (4.3)
*Enterobacter spp.*	19 (7.8)	8 (8.4)	2 (4.3)
*Serratia marcescens*	14 (5.7)	0 (0)	4 (8.7)
*Acinetobacter baumannii*	15 (6.1)	1 (1.1)	2 (4.3)
*Stenotrophomonas maltophilia*	6 (2.4)	6 (6.3)	2 (4.3)
Polymicrobial microorganisms	62 (25.3)	31 (32.3)	9 (18.5)
Two Gram-positive cocci	2 (0.8)	0 (0)	0 (0)
Two Gram-negative bacilli	22 (9.0)	10 (10.4)	3 (6.5)
Combined Gram-positive and Gram-negative	37 (15.1)	20 (20.8)	6 (13.0)
≥3 microorganisms	1 (0.4)	1 (1.1)	0 (0)
Others *	10 (4.1)	3 (3.1)	0 (0)
Normal flora or no growth	7 (2.9)	0 (0)	2 (4.3)

* Including *Burkholderia cepacia* (3), *Corynebacterium striatum* (3), *Morganella* species (1), *Citrobacter koseri* (1), *Moraxella catarrhalis* (1), and *Hemophilus influenzae* (1).

**Table 2 antibiotics-09-00760-t002:** Patient demographics, characteristics, and clinical presentation of all neonatal ventilator-associated pneumonia (VAP) at CGMH, October 2017 to March 2020.

Characteristics	All VAP Episodes (Total *n* = 245)	MDR Pathogen-Associated VAP Episodes (Total *n* = 96)	Non-MDR Pathogen-Associated VAP Episodes (Total *n* = 149)	*p* Values
Cases Demographics				
Gestational age (weeks), median (IQR)	26.0 (25.0-28.0)	26.0 (25.0-28.8)	26.0 (25.0-28.0)	0.770
Birth weight (g), median (IQR)	876.0 (725.0-1092.5)	907.5 (746.0-1101.5)	850 (700-1091.5)	0.553
Gender (male/female), *n* (%)	147 (60.0)/98 (40.0)	59 (61.5)/37 (38.5)	88 (59.0)/61 (41.0)	0.790
5 minutes Apgar score ≤7, *n* (%)	79 (32.2)	32 (33.3)	47 (31.5)	0.771
Inborn/outborn, n (%)	213 (86.9)/32 (13.1)	82 (85.4)/14 (14.6)	131 (87.9)/18 (12.1)	0.567
Birth by NSD/Cesarean section, *n* (%)	96 (39.2)/149 (60.8)	45 (46.9)/51 (53.1)	51 (34.2)/98 (65.8)	0.060
Respiratory distress syndrome (≥Gr II), *n* (%)	169 (69.0)	59 (61.5)	110 (73.8)	0.048
Intraventricular hemorrhage (≥Stage III), *n* (%)	19 (7.7)	7 (7.3)	12 (8.1)	0.646
Underlying Chronic Comorbidities, *n* (%)				
Neurological sequelae	81 (33.1)	33 (34.4)	48 (32.2)	0.781
Bronchopulmonary dysplasia	164 (66.9)	60 (62.5)	104 (69.8)	0.328
Cardiovascular diseases	35 (14.3)	13 (13.5)	22 (14.8)	0.854
Gastrointestinal sequelae	70 (28.6)	30 (31.3)	40 (26.8)	0.469
Renal disorders	6 (2.4)	4 (4.2)	2 (1.3)	0.158
Congenital anomalies	21 (8.6)	8 (8.3)	13 (8.7)	0.922
Presences of any chronic comorbidities	203 (82.9)	77 (80.2)	126 (84.6)	0.570
Presences of more than one comorbidities	108 (44.1)	43 (44.8)	65 (43.6)	0.741
Day of life at onset of VAP (day), median (IQR)	29.0 (19.0–48.0)	26.5 (15.5–50.8)	30.0 (20.0–46.5)	0.377
On antibiotic treatment at onset of VAP, *n* (%)	88 (35.9)	39 (40.6)	49 (32.9)	0.223
Use of TPN and/or intrafat, *n* (%)	182 (74.3)	73 (76.0)	109 (73.2)	0.655
Use of central venous catheter, *n* (%)	225 (91.8)	92 (95.8)	133 (89.3)	0.093
Clinical Features, *n* (%)				
Fever	14 (5.7)	7 (7.3)	7 (4.7)	0.280
On HFOV/conventional ventilator	89 (36.3)/156 (63.7)	37 (38.5)/59 (61.5)	59 (39.6)/97 (65.1)	0.588
Septic shock	39 (15.9)	17 (17.7)	22 (14.8)	0.593
Metabolic acidosis	39 (15.9)	12 (12.5)	27 (18.1)	0.285
NTISS score at onset of VAP, median (IQR)	27.0 (25.0-29.0)	27.8 (26.0-29.8)	27.0 (24.0-29.0)	0.313
With concurrent bacteremia	46 (18.8)	23 (24.0)	23 (15.4)	0.131
Requirement of blood transfusion *	180 (73.5)	73 (76.0)	107 (71.8)	0.456
Requirement of high FiO_2_ (≥50%) ^#^	99 (40.4)	45 (46.9)	54 (36.2)	0.110
Chest X-ray findings				0.777
New infiltrate	98 (40.0)	40 (41.7)	58 (38.9)	
Worsening infiltrate	134 (54.7)	52 (54.2)	82 (55.0)	
Persistent infiltrate	13 (5.3)	4 (4.2)	9 (6.0)	

NSD: normal spontaneous delivery; IQR: interquartile range; HFOV: high-frequency oscillatory ventilator; NTISS score: Neonatal Therapeutic Intervention Scoring System; TPN: total parenteral nutrition. * Including leukocyte poor red blood cell and/or platelet transfusion. # To maintain SpO_2_ (pulse oximetry) >94%.

**Table 3 antibiotics-09-00760-t003:** Therapeutic intervention and outcomes of all neonatal ventilator-associated pneumonia (VAP) at CGMH, October 2017 to March 2020.

Characteristics	All VAP Episodes (Total *n* = 245)	MDR Pathogens Associated VAP Episodes (Total *n* = 96)	Non-MDR Pathogens Associated VAP Episodes (Total *n* = 149)	*p* Values
Therapeutic Intervention, *n* (%)				
Initial empiric antibiotics				
Inappropriate initial antibiotics	56 (22.9)	49 (51.0)	7 (4.7)	<0.001
Use of first line antibiotics	50 (20.4)	59 (61.5)	88 (59.0)	0.070
Use of broad-spectrum antibiotics	195 (79.6)	32 (33.3)	47 (31.5)	0.070
Modification of therapeutic antibiotics	156 (63.7)	74 (77.1)	82 (55.0)	<0.001
Therapeutic antibiotics				
Use of first line antibiotics	74 (30.2)	18 (18.8)	56 (37.6)	0.002
Use of broad-spectrum antibiotics	171 (69.8)	78 (81.3)	93 (62.4)	0.002
Duration of antibiotic treatment (day), mean ± SD	10.5 ± 3.8	11.0 ± 3.9	9.9 ± 3.0	0.010
Therapeutic Outcomes, *n* (%)				
Detailed clinical assessment, *n* (%)				0.064
Clinical resolution	96 (39.2)	26 (27.1)	70 (47.0)	<0.001
Delayed resolution	75 (30.6)	37 (38.5)	38 (25.5)	0.034
Relapse or recurrent infection	21 (8.6)	12 (12.5)	9 (6.2)	
Superinfection	42 (17.1)	17 (17.7)	25 (16.8)	
Death	11 (4.5)	4 (4.2)	7 (4.7)	
Detailed microbial assessment, *n* (%)				0.256
Resolution	114 (46.5)	42 (43.8)	73 (49.0)	
Relapsed or recurrent infection	36 (5.7)	20 (20.8)	16 (10.7)	
Superinfection	77 (31.4)	28 (29.2)	49 (32.9)	
Clinical failure	18 (15.9)	6 (6.3)	12 (8.1)	
Overall clinical assessment, *n* (%)				0.920
Cure	206 (84.1)	81 (84.4)	125 (83.9)	
Treatment failure *	39 (15.9)	15 (15.6)	24 (16.1)	

IQR: interquartile range. * Treatment failure was defined as neonates who required therapeutic antibiotics for more than two weeks, those that progressed to bacteremia, those with worsening clinical symptoms after appropriate antibiotics for one week, and neonates who died due to ventilator-associated pneumonia.

**Table 4 antibiotics-09-00760-t004:** Multivariate logistic regression analysis for independent risk factors of clinical treatment failure in neonatal ventilator-associated pneumonia.

Variables	Univariate Analysis		Multivariate Analysis	
	OR (95% CI)	*p*-Values	Adjusted OR (95% CI)	*p*-Values
Gestational Age				
<26 weeks	1.09 (0.39–3.23)	0.875		
26–28 weeks	1.11 (0.36–3.40)	0.849		
29–33 weeks	1 (reference)			
≥34 weeks	1.67 (0.42–6.56)	0.465		
Septic shock	4.05 (1.87–8.81)	<0.001	3.06 (1.07–8.72)	0.037
On HFOV vs. conventional ventilator	4.54 (2.19–9.41)	<0.001	4.10 (1.70–9.88)	0.002
Inappropriate initial antibiotics	0.70 (0.29–1.69)	0.428		
MDR pathogens associated VAP	0.97 (0.48–1.95)	0.920		
Presences of neurological sequelae	5.49 (2.64–11.44)	<0.001	3.35 (1.47–7.67)	0.004
Bronchopulmonary dysplasia	0.65 (0.32–1.32)	0.234		
Severity of Illness at Onset of VAP				
Every 3 increase in NTISS scores	1.65 (1.16–2.36)	0.006	0.91 (0.55–1.50)	0.716
Concurrent sepsis	4.72 (2.24–9.93)	<0.001	4.83 (2.03–11.51)	<0.001
Thrombocytopenia	2.79 (1.16–6.70)	0.022	1.53 (0.54–4.35)	0.430

HFOV: high-frequency oscillatory ventilator; OR: odds ratio; 95% CI: 95% confidence interval; MDR: multidrug-resistant; NTISS: Neonatal Therapeutic Intervention Scoring System.

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
