# Peer review of "Impacts of Multidrug-Resistant Pathogens and Inappropriate Initial Antibiotic Therapy on the Outcomes of Neonates with Ventilator-Associated Pneumonia"

_antibiotics, 2020, doi:10.3390/antibiotics9110760_

Round 1

Reviewer 1 Report

The manuscript is well written, the results are well presented and sufficiently discussed. Even though the significant effect of MDR pathogens on the outcomes has not been found, it is emphasized that further investigation of the issue and developing a stewardship program is urgently needed.

I have just a couple of technical comments:

It might be worth including some information about MDR in the introduction: prevalence in particular settings etc.

Please use the uniformed abbreviations (for example, NBBAL, line 77 or NB-BAL, line 131).

There are many abbreviations in the text which makes it difficult to read sometimes. Please consider avoiding abbreviations that are used only once-twice in the text.

Line 233: “does” instead of “dose”.

Author Response

RE: Antibiotics-977131

Impacts of multidrug-resistant pathogens and inappropriate initial antibiotic therapy on the outcomes of neonates with ventilator-associated pneumonia

Dear Editor,

Thank you for your appreciated comments on our manuscript. We had the manuscript revised, all according to the reviewers’ and editor’s suggestions. We underline every change and highlight in red color on the revised manuscript. The replies for the reviewers’ criticisms are as followings. We hope this revised version can be acceptable. 

Best regards,

Ming-Horng Tsai

Chief, Division of Neonatology and Pediatric Hematology/Oncology, Department of Pediatrics, Yunlin Chang Gung Memorial Hospital, Taiwan, R.O.C.

Comments from Reviewer No.1 :

The manuscript is well written, the results are well presented and sufficiently discussed. Even though the significant effect of MDR pathogens on the outcomes has not been found, it is emphasized that further investigation of the issue and developing a stewardship program is urgently needed.

I have just a couple of technical comments:

It might be worth including some information about MDR in the introduction: prevalence in particular settings etc.

Reply:

Thank you for your appreciated comments and instructive advice. I will include some information about MDR in the introduction. In the second paragraph of the introduction as following: Furthermore, healthcare associated infections (HAIs) caused by multidrug-resistant (MDR) pathogens are becoming alarmingly frequent in the NICU, accounting for nearly one-fifth of all nosocomial infections, and are associated with higher mortality and morbidity [18-20].

Please use the uniformed abbreviations (for example, NBBAL, line 77 or NB-BAL, line 131).

Reply:

Thank you for your instructive advice. I will use the uniformed abbreviations, thank you. I revise all NB-BAL in the text to be NBBAL, to make it uniformed, thank you.

There are many abbreviations in the text which makes it difficult to read sometimes. Please consider avoiding abbreviations that are used only once-twice in the text.

Reply:

     Thank you for your instructive advice. I will avoid abbreviations that are used only once-twice in the text, thank you. I have cancelled the abbreviations including IVH, PVL, AAP, NEC, and IDSA accordingly, thank you.

Line 233: “does” instead of “dose”.

Reply:

     Thank you for your instructive advice. I will revise it as does accordingly, thank you.

Reviewer 2 Report

The abstract is well structured, organized and contains the necessary information to understand this article.

Is there any specific reason why the line space in the text is so narrow, unlike the references list?

The introduction should provide detailed information about pathogens (bacteria and or fungi) associated with ventilator-associated pneumonia (VAP), bacteremia and or septic shock in the neonate, the commonly found drug resistance and or mechanisms with these bacteria and explain why the antibiotic misuse or overuse can be of high risk.

The author can also detail how inappropriate empiric treatment can affect the early colonization in the newborn.

It will give a clearer picture of the work and let the reader understand the implications and the consequences

Line 64-65: is Pseudomonas, the only genus colonization affected? By the way, you have not specified any species so change "species" to "genus" in the text.

Table1: how does the author define the limit between microorganism early colonization and infection? Are CFU taken into consideration?

Why does the author think Candida albicans isolated from respiratory tract specimen are colonization microorganisms and not cause of infection?

Can the author provide details about the method of detection of antimicrobial resistance? Standard diffusion method? Molecular method (PCR)? Etc …  as well as antimicrobial resistance profile of each microorganism  and how multidrug resistance is defined according to their data in a supplementary material

This information along with the empiric drug used can enable to interpret this data correctly

It is not easy to read the description of the results and follow data in tables. The results section is not well structured and is not exhaustively detailed.

Eg:

  1. Case prevalence
  2. Pathogens and antimicrobial resistance profile
  3. Etc…

It all depends on the story the author wanted to bring out is the discussion.

I am still confused about the conclusion of the study

The title is starting with “Impacts ….” And at the end, the author stated that “Inappropriate initial antibiotic  treatment did not significantly affect the outcomes.” Am I missing the point?

Author Response

RE: Antibiotics-977131

Impacts of multidrug-resistant pathogens and inappropriate initial antibiotic therapy on the outcomes of neonates with ventilator-associated pneumonia

Dear Editor,

Thank you for your appreciated comments on our manuscript. We had the manuscript revised, all according to the reviewers’ and editor’s suggestions. We underline every change and highlight in red color on the revised manuscript. The replies for the reviewers’ criticisms are as followings. We hope this revised version can be acceptable. 

Best regards,

Ming-Horng Tsai

Chief, Division of Neonatology and Pediatric Hematology/Oncology, Department of Pediatrics, Yunlin Chang Gung Memorial Hospital, Taiwan, R.O.C.

Comments from Reviewer No.2:

The abstract is well structured, organized and contains the necessary information to understand this article. Is there any specific reason why the line space in the text is so narrow, unlike the references list?

Reply:

Thank you for your appreciated comments and question. In my manuscript, the line spaces are all uniformed in the text and in the references list (all double space). I think it is the format change after I submitted this manuscript, where the system converted my manuscript to so narrow line space in the text, unlike the references list. I think I will discuss with the editorial manager about this issue, thank you.

The introduction should provide detailed information about pathogens (bacteria and or fungi) associated with ventilator-associated pneumonia (VAP), bacteremia and or septic shock in the neonate, the commonly found drug resistance and or mechanisms with these bacteria and explain why the antibiotic misuse or overuse can be of high risk.

Reply:

Thank you for your instructive advice. I will provide detailed information in the end of the first paragraph of the introduction as following: In contrast to coagulase-negative Staphylococcus (CoNS) being the most common cause of neonatal late-onset sepsis, Gram-negative bacilli account for the majority of neonatal VAP [6-9]. Because clinicians tend to use broad-spectrum antibiotics when critically ill neonates have clinical deterioration [8,9], these patients are likely to have antibiotic overuse. Therefore, antibiotic-resistant pathogens are emerging after antibiotic selection and neonates face an increased risk of antibiotic-associated composite outcome [10-12]. Therefore, I provide information about pathogens associated VAP and bacteremia, the commonly found drug resistance mechanisms and explain why antibiotic misuse or overuse can be of high risk.

The author can also detail how inappropriate empiric treatment can affect the early colonization in the newborn. It will give a clearer picture of the work and let the reader understand the implications and the consequences

Reply:

     Thank you for your instructive advice. I have tried to detail how inappropriate empiric treatment can affect the early colonization in the newborn. However, I can’t find any previous studies that investigated how inappropriate empiric treatment can affect the early colonization in the newborn. I found treatment outcomes, subsequent emerging increasing antimicrobial resistance, and infectious complications are the study end points of most inappropriate empiric treatment in all settings. It is also difficult to define “early colonization” in the newborn. Maybe in the future this issue can be investigated by enrolling all preterm infants who receive empiric antibiotics since birth and checking their colonization before 2 weeks old as the study end points. I am sorry that I can’t detail how inappropriate empiric treatment can affect the early colonization in the newborn. If the reviewer insists, please give me more time to think about this issue.

Line 64-65: is Pseudomonas, the only genus colonization affected? By the way, you have not specified any species so change "species" to "genus" in the text.

Reply:

     Thank you for your instructive advice. I will change “species” to “genus” in the text of line 64-65, thank you.

Table 1: how does the author define the limit between microorganism early colonization and infection? Are CFU taken into consideration?

Reply:

     Thank you for your question. We define the limit between microorganism early colonization and infection based on the updated CDC criteria for neonatal VAP; please see the appendix table 1. Because the CDC permits the diagnosis of “Clinically defined pneumonia” based only on clinical and radiological findings without any isolated pathogen (please see references no.1 and no. 23), the CFU is not taken into consideration. I have mentioned this issue in the definition section. (originally line 91-93).

Why does the author think Candida albicans isolated from respiratory tract specimen are colonization microorganisms and not cause of infection?

Reply:

     Thank you for your question. It is extremely rare in the NICU that Candida species actually cause VAP. There has been study that investigated whether Candida spp. cause VAP or are only colonization (published in Clin Microbiol Infect 2016;22(1):94.e1-94.e8). They concluded that Candida spp. isolation in respiratory samples are mostly colonization. Only Candida spp. isolated from pleural fluid culture will be the true pathogens. Therefore, we followed this rule and none of the Candida spp. isolated from respiratory specimen was treated, and finally proved to be the colonization microorganisms. 

Can the author provide details about the method of detection of antimicrobial resistance? Standard diffusion method? Molecular method (PCR)? Etc …  as well as antimicrobial resistance profile of each microorganism and how multidrug resistance is defined according to their data in a supplementary material

Reply:

     Thank you for your instructive advice. In the method section, I have provided details about detection of antimicrobial resistance and how multidrug resistance is defined according to our data as following: Antibiotic susceptibility patterns were determined according to methods recommended by the National Committee for Clinical Laboratory Standards Institute (CLSI) for disk diffusion method and categorical assignment was performed using CLSI breakpoints [25]. Multidrug-resistant (MDR) pathogens were defined as those resistant to at least one agent in three or more antimicrobial categories: carbapenems (imipenem and meropenem); penicillins (piperacillin, ticarcillin and piperacillin/tazobactam); broad-spectrum cephalosporins (ceftazidime and cefepime); monobactams (aztreonam); aminoglycosides; and fluorquinolones [20,26]. (line 98-line 104 of the original manuscript). 

For the antimicrobial resistance profile of each microorganism, it is quite complicated to list all the antimicrobial susceptibility to all 9-12 antibiotics for nearly 20 microorganisms. Besides, we checked whether these microorganisms are multidrug resistant or not just through some “key antibiotics”. For example, we considered Pseudomonas as MDR or non-MDR based on the susceptibility to gentamicin and ceftazidime. Therefore, it takes more time to list the antimicrobial resistance profile of each microorganism in a supplementary material. If the reviewer insists, please let me know and give us more time, thank you.

This information along with the empiric drug used can enable to interpret this data correctly

It is not easy to read the description of the results and follow data in tables. The results section is not well structured and is not exhaustively detailed.

Eg:

1 Case prevalence

2 Pathogens and antimicrobial resistance profile

3 Etc…

Reply:

Thank you for your questions and instructive advice. Actually we first described the epidemiology and microorganisms of VAP in our cohort. Secondary we compared MDR VAP and non-MDR VAP. Then we investigated whether inappropriate initial antibiotics would affect the outcomes and finally, we investigated the independent risk factors of treatment failure. I will add the subheadings in the result section as following:

Epidemiology of VAP and the microorganisms

Comparisons between MDR VAP and non-MDR VAP

Therapeutic outcomes and impacts of inappropriate initial antibiotics

Please follow up the table and subheadings in the results section, thank you.

It all depends on the story the author wanted to bring out is the discussion.

I am still confused about the conclusion of the study

The title is starting with “Impacts ….” And at the end, the author stated that “Inappropriate initial antibiotic treatment did not significantly affect the outcomes.” Am I missing the point?

Reply:

     Thank you for your question. The title is starting with “Impacts….” because we aimed to investigate the impacts of MDR pathogens and inappropriate initial antibiotic treatment on the outcomes” However, after our investigation, we found that “Inappropriate initial antibiotic treatment did not significantly affect the outcomes.” It is completely logical after our analyses and multivariate logistic regression. If we found inappropriate initial antibiotic treatment did affect the outcomes, we will have it in the conclusion. So, you are not missing any point. 
